# Adaptive Pruning of Channel Spatial Dependability in Convolutional Neural Networks

## ABSTRACT

Deep Convolutional Neural Networks (CNNs) have demonstrated excellent performance in various multimedia application scenarios. However, complex models often require significant computational resources and energy costs. Therefore, CNN compression is crucial for addressing deployment challenges of multimedia application on resource constrained edge devices. However, existing CNN channel pruning strategies primarily focus on the "weights" or "activations" of the model, overlooking its "interpretability" information. In this paper, we explore CNN pruning strategies from the perspective of model interpretability. We model the correspondence between channel feature maps and interpretable visual perception based on class saliency maps, aiming to assess the contribution of each channel to the desired output. Additionally, we utilize Discrete Wavelet Transform (DWT) to capture the global features and structure of class saliency maps. Based on this, we propose a Channel Spatial Dependability (CSD) metric, evaluating the importance and contribution of channels in a bidirectional manner to guide model quantization pruning. And we dynamically adjust the pruning rate of each layer based on performance changes, in order to achieve more accurate and efficient adaptive pruning. Experimental results demonstrate that our method achieves significant results across a range of different networks and datasets. For instance, we achieved a 51.3% pruning on the ResNet-56 model while maintaining an accuracy of 94.16%, outperforming feature-map or weight-based pruning and other State-of-the-Art (SOTA).

## CCS CONCEPTS

• **Computing methodologies** → **Computer vision**; **Neural networks**.

## KEYWORDS

Model Compression, Deep Neural Networks, Channel Pruning, Class Saliency Maps

## 1 INTRODUCTION

Convolutional Neural Networks (CNNs) have been widely applied in various computer vision tasks, including image classification [38, 47–49], semantic segmentation [4, 12, 20], object detection [3, 13, 15], and many more challenging tasks. However, for increasingly complex tasks, while increasing the depth of CNNs to enhance

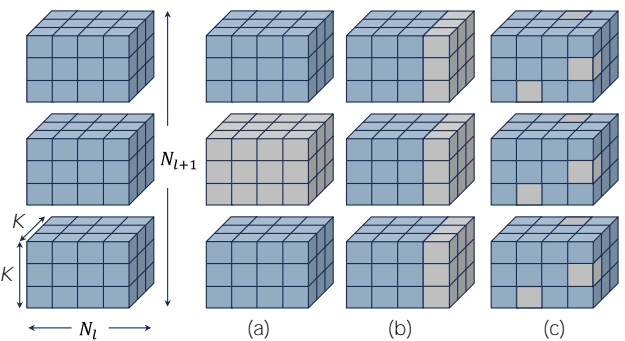

**Figure 1: In the context of different pruning granularities, where the input dimension is represented as $N_l$, the output dimension as $N_{l+1}$, and the kernel size as $K^2$, (a) *filter-wise* and (b) *channel-wise* both constitute structured pruning methods. Conversely, (c) *group-wise* is non-structured pruning involves grouping weights along the output dimension $N_{l+1}$.**

their feature representation capability, the models become excessively parameterized, making it challenging to deploy them on resource-constrained embedded devices. Therefore, effectively reducing the parameter count and floating-point operations of CNN models while ensuring that their performance does not significantly degrade is crucial for the practical application of deep learning technologies. To address this challenge, various model compression strategies have emerged, including network pruning [2, 10, 41], parameter quantization [29, 34, 35], low-rank approximation [1, 27, 40], knowledge distillation [54, 58, 59] and others.

Network pruning is an effective approach for reducing the parameter count and decreasing the computational workload of CNNs, with broad prospects for applications. Typically, based on whether the structure of the network changes before and after pruning, network pruning can be divided into two categories: structured pruning [18, 23, 31, 56] and unstructured (weight) pruning [11, 28, 36], as shown in **Figure 1**. Unstructured pruning usually refers to fine-grained pruning methods with relatively high pruning accuracy. Structured pruning typically takes channels or filters in convolutional layers as the basic pruning units, preserving the model's structure. While the former can achieve higher pruning rates, it cannot leverage general-purpose hardware for acceleration, whereas the latter can be accelerated using commonly available hardware.

In this paper, we focus on structured pruning, where the core challenge lies in reducing the number of intermediate features. Classic methods involve evaluating the importance of channels and pruning under certain constraints to make the model sparse. Traditional evaluation methods include norms-based [26], geometric median-based [18], and Hessian-based [55] approaches. Although these methods can compress the model, they may not fully capture

the contribution of channels to the overall performance of the model. Norms-based methods may not provide a complete understanding of feature and gradient information, potentially overlooking important channels or retaining some less important ones. Geometric median-based methods may lack flexibility when dealing with complex datasets or large-scale models, making it challenging to adapt to various data distributions and model structures, resulting in inaccurate evaluations. Hessian-based methods typically require computing higher-order derivative information, which increases computational costs. Additionally, Hessian methods may encounter issues with local optima and computational stability in non-convex optimization problems. In summary, while these evaluations provide some guidance for channel pruning.

Recent findings have demonstrated that layer-wise adaptive sparsity [9, 25, 50] is a superior pruning approach. However, these methods only consider existing evaluation criteria, as shown in **Figure 2**. Inspired by these studies, we propose layer-wise adaptive compression method based on a novel filter metric standards. Our goal is to develop an effective and efficient compression method that identifies the most valuable channels in the network under performance loss constraints. We integrate forward feature information and gradient information from specific class backpropagation to extract high-level semantic information for a given task. Additionally, we employ novel metrics, such as Channel Spatial Dependability (CSD), constructed using other methods like Discrete Wavelet Transform (DWT). Unlike previous approaches, we not only focus on filter evaluation criteria but also integrate them with layer-wise adaptive sparsity methods. We conduct tests on CIFAR-10 [24], CIFAR-100 [24] and ImageNet [42] datasets across various architectures. Furthermore, we perform comprehensive tests through ablation studies to examine the robustness of our method. Our experimental results demonstrate competitive performance against current state-of-the-art (SOTA) benchmarks [23, 30, 31, 52]. We summarize the main contributions of our paper as follows:

- In order to better understand the internal structure of CNNs, we propose the CSD evaluation criterion. By assessing the spatial dependability of each channel, we can determine which channels carry crucial information for model decisions, enabling effective pruning.
- We integrate the CSD with layer-wise adaptive sparsity algorithms to more accurately determine the optimal number of filters for each layer of the model. This makes the pruning process more intelligent and efficient, preserving model performance stability and accuracy while reducing the number of model parameters.
- We validate our algorithm on datasets such as CIFAR-10, CIFAR-100, and ImageNet, achieving state-of-the-art performance. Through extensive experimentation, we demonstrate the effectiveness and reliability of our proposed method across different datasets.

## 2 RELATED WORK

### 2.1 Structed Pruning

Structured pruning typically prunes channels or filters as the basic units in convolutional layers, employing two main approaches: one based on the filters themselves and the other based on feature maps.

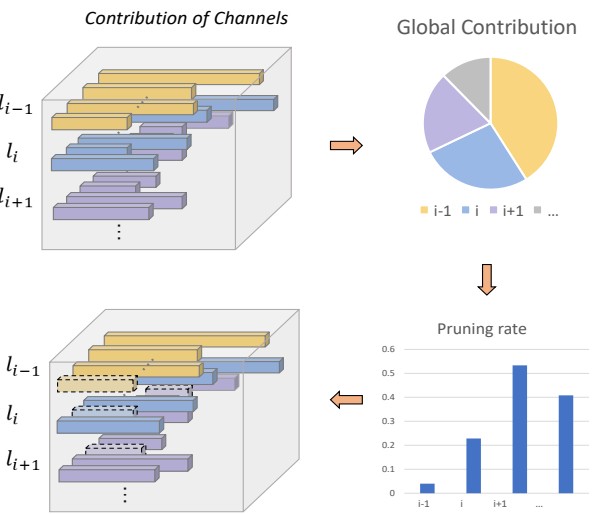

**Figure 2: Adaptive layer pruning rate method. This mechanism automatically calculates the pruning rate for each layer by considering the importance of the channel or filter (according to a certain evaluation criterion, the evaluation criterion in this article is CSD) and the overall pruning rate, thereby reducing the need for manual intervention.**

Determining their saliency is a critical step in model compression techniques, and various approaches exist to address this challenge, each with its own advantages and limitations. Noteworthy methods using filter-centric approaches include [26], who suggest sorting based on the L1 norm of convolutional kernels, considering smaller norms as less important and eligible for pruning. Additionally, approaches such as those by [18] utilize the Geometric Median for model pruning, replacing filters that are deemed too similar to others. Molchanov *et al.* [39] estimate filter significance by prioritizing the ranking of first-order Taylor coefficients. On the other hand, methods relying on feature maps for judgment, such as the one proposed by Lin *et al.* [31], determine filter importance based on the rank of feature maps. Sui *et al.* [46] suggest using channel independence to measure the correlation between different feature maps, thereby enabling effective filter pruning. Among structured pruning techniques, channel pruning is particularly popular as it operates at a finer level compared to filter pruning, making it well-suited for deep learning frameworks, as shown in **Figure 1**.

### 2.2 Interpretation Methods

Due to the overparameterized nature of CNNs, involving millions of parameters stacked over hundreds of layers, the prediction results of deep models are often challenging to interpret [43]. Several interpretability tools [22, 44] have been proposed to explain or reveal the decision-making process of deep models. For a given input sample, the model's output is calculated relative to the input feature maps using the backpropagation algorithm, computing the gradients of the model output with respect to each pixel in the input feature maps. These gradients indicate the influence of each pixel in the input feature maps on the output. Transforming these gradients

Figure 3: Overview of CSD(Channel Spatial Dependability). CSD is a method aimed at finding the optimal pruning strategy to enhance interpretability for each convolutional layer. This approach transforms the problem into a global pruning optimization problem, by calculating the contribution of each channel to predictions (by analyzing the class activation maps corresponding to each channel), and utilizing discrete wavelet transformation to preserve low-frequency components as the criterion for evaluating filter importance. Under local constraints, this method aims to maximize the compression ratio of each convolutional layer $n^l$, while dynamically searching for the optimal pruning rate for that layer based on performance loss. This method provides an adaptive filter pruning strategy based on model interpretability.

into importance or relevance maps of the input feature maps can be achieved through post-processing or transformations such as taking absolute values, squaring or other operations to represent the impact of features on the output.

Visualizing these importance or relevance maps provides a basis for interpreting the model's decisions. These images can reveal which input features play a crucial role in the model's output, offering some explanation for the model's decision. By visualizing feature map gradients or employing other methods, researchers and practitioners can better understand how the model makes predictions or decisions based on input features. Such explanations contribute to increasing the transparency of the model and aiding users in gaining a deeper understanding of the model's behavior.

## 2.3 Discussion

Based on extensive research, we have identified a significant advantage in measuring channel importance based on feature maps. However, most of these methods primarily focus on feature map information while neglecting interpretability. Therefore, we aim to leverage the interpretability of the model to provide more comprehensive guidance for a more efficient pruning process. In addition, traditional pruning methods often require extensive hyperparameter tuning and complex training, leading to substantial human resource costs. We propose an automatic method to calculate the pruning rate for each layer, aiming to achieve more efficient acceleration and achieve stronger performance.

## 3 METHOD

### 3.1 Notations

Let's assume a standard CNN model with $L$ convolutional layers, indexed by $l \in \{0, \ldots, L-1\}$. For the $l$-th convolutional layer, which contains $n^l$ filters $F_i^l \in \mathbb{R}^{n^{l-1} \times k^l \times k^l}$, we can compute the number of parameters for this layer as $W^l = \{F_1^l, F_2^l, ..., F_{n^l}^l\} \in \mathbb{R}^{n^l \times n^{l-1} \times k^l \times k^l}$, where $n^l$, $n^{l-1}$, and $k^l$ represent the number of output channels, number of input channels, and kernel size for the $l$-th layer. The total number of parameters for all convolutional layers in the entire network can be denoted as $T = \sum_{l=0}^{L-1} W^l$.

We introduce a new symbol $p^l$ to represent the pruning rate for the $l$-th layer, where $p^l$ ranges between 0 and 1. With this, we can calculate the new number of filters for this layer as $n_{\text{new}}^l = \lceil p^l \times n^l \rceil$. Consequently, the parameter quantity for this layer is given by $W_{\text{new}}^l \in \mathbb{R}^{n_{\text{new}}^l \times n_{\text{new}}^{l-1} \times k^l \times k^l}$. Similarly, after pruning, the total parameters for the CNNs convolutional layers can be expressed as $T_{\text{new}} = \sum_0^{L-1} W_{\text{new}}^l$. Therefore, network pruning can be formulated as the following optimization problem:

$$\min_{\{W_i^l\}_{i=1}} \mathcal{L}(y, f(X, W^l)), s.t. \qquad T_{\text{new}} \leq T \times P_t, \qquad (1)$$

where $\mathcal{L}(y, f(X, W^l))$ represents the loss function, $y$ denotes the ground truth labels, $X$ is the input data, $f(X, W^l)$ is the CNN model's output function with parameters $\{W_i^l\}_{i=1}$, $T \times P_t$ denotes the desired size of the pruned model's parameters, and $P_t$ is the targeted total pruning rate.

Over time, channel importance has been predominantly based on feature maps. However, feature maps do not comprehensively reflect the model's understanding of the data and fail to capture the dynamic changes during model training. Considering that gradients indicate the update direction and importance of each parameter in the model, they not only express the model's sensitivity to each parameter but also reflect the model's attention to different features during training. Therefore, we propose gradient-enhanced feature maps $C_i^l$, which complement the limitations of feature map evaluation by utilizing gradient information. This approach allows for a comprehensive consideration of the model's understanding of the data and its attention to features during training. This combination enables a more accurate assessment of each channel's contribution to model performance, facilitating the selective retention of channels crucial for model decision-making and further optimizing model performance and efficiency, as shown in **Figure 3**.

Firstly, we conduct forward propagation through the CNNs to obtain feature maps $F_i^l$. Subsequently, we compute the score list $s$ for all classes. Assuming there are $K$ classes, we set the score corresponding to the correct class index $c$ to 1 and the rest to 0, yielding the score $S_c$ for the specific class $c$:

$$S_c = [s_1, s_2, \ldots, s_K] \; s.t. \quad s_k = \delta_{kc} = \begin{cases} 1 & \text{if } k = c \\ 0 & \text{otherwise} \end{cases}, \qquad (2)$$

Subsequently, we utilize $S_c$ to backpropagate and obtain gradient information, denoted as grad:

$$grad = \nabla_{\text{inputs}}(\text{pred} \odot S_c), \qquad (3)$$

where $\nabla_{\text{inputs}}$ represents the gradient of the model output with respect to the inputs, and $\odot$ denotes element-wise multiplication. This allows us to track the class saliency information $w_c$ of the $i$-th filter in the $l$-th convolutional layer for a specific class $c$:

$$w_c = \frac{1}{H \times W} \sum_{i=1}^H \sum_{j=1}^W \text{grad}_{(i,j)}, \qquad (4)$$

where $H$ and $W$ represent the height and width of the feature map, respectively, and $\text{grad}_{(i,j)}$ denotes the gradient value at position $(i, j)$ on the feature map, obtained during model backpropagation.

When attempting to interpret model decisions, it is crucial to understand which channels play a critical role in the model's final predictions. We expect convolutional layers' channels to strike the optimal balance between feature analysis and sensitivity analysis. Feature maps are commonly utilized for feature analysis and extraction, aiding in understanding how networks gradually extract abstract and useful features to accomplish specific tasks. Gradient information flowing into CNNs assigns importance values to each neuron for specific decisions. Combining these aspects allows us to visualize the image regions on which the model's predictions for specific classes rely, facilitating the interpretation of the model's decision-making process. This enhances the model's interpretability and aids in verifying whether the model makes predictions based on reasonable features.

Therefore, we propose utilizing gradient-enhanced feature maps $C_i^l$ to better understand the CNNs internal structure:

$$C_i^l = w_c \times F_i^l, \qquad (5)$$

where $F_i^l$ denotes the feature map of the $i$-th channel in the $l$-th convolutional layer, and $w_c$ represents the gradient information for a specific class in the feature map. Multiply these gradients with their corresponding feature maps element-wise to abtain gradient enhanced feature maps.

## 3.2 CSD Criterion

If the feature information is directly fused with gradient information, it may lead to excessive redundancy or blurring, making it difficult to distinguish the truly beneficial parts for decision-making. To address this issue, we propose a novel filter evaluation criteria—CSD, inspired by the interpretability of the model and Discrete Wavelet Transform (DWT), as shown in **Figure 3**. DWT can decompose an image into frequency components at different scales, where high-frequency components often contain noise, and extracting the low-frequency components containing the overall trend and basic structural information of the signal helps reduce redundant information. Therefore, we use DWT to decompose the gradient-enhanced feature map $C_i^l$ from Equations (2)-(5) into low-frequency and high-frequency parts:

$$\text{Dec}\left(C_i^l\right) \rightarrow \begin{cases} C_i^l \circledast L \\ C_i^l \circledast H \end{cases}$$
$$= \left(\mathcal{F}_{\text{Low}}\left(C_i^l\right), \mathcal{F}_{\text{High}}\left(C_i^l\right)\right), \qquad (6)$$

where Dec represents the decomposition process, $L$ and $H$ represent the low-pass and high-pass filters used for DWT, and $\mathcal{F}_{\text{Low}}$ and $\mathcal{F}_{\text{High}}$ represent the low-frequency and high-frequency parts of the gradient-enhanced feature map obtained through wavelet decomposition. The high-frequency part typically contains image details, while the low-frequency part contains the overall structure and global information of the image. Retaining only the low-frequency part of the gradient-enhanced feature map allows for aggregating global information and better capturing its overall characteristics:

$$\text{Rec}\left(\mathcal{F}_{\text{Low}}\left(C_i^l\right), 0\right) = C_i^l$$
$$= \mathcal{F}_{\text{Low}}\left(C_i^l\right) \circledast \tilde{L} + \mathcal{F}_{\text{High}}\left(C_i^l\right) \circledast 0, \qquad (7)$$

where Rec represents the reconstruction process, $\tilde{L}$ represents the conjugate filter of $L$. We choose the Coiflet wavelet for its good compact support, helping to preserve more information of the original signal in wavelet transform. Additionally, considering that the

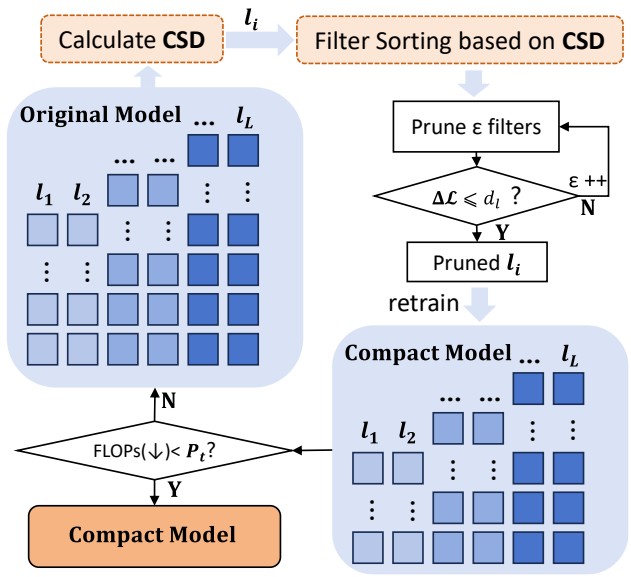

**Figure 4: Schematic diagram of adaptive pruning rate for each layer. Randomly input a certain number of samples into the original model, calculate the CSD for each channel. For each layer, prune the $k_l$ channel with the minimum CSD (which needs to be initialized), and calculate the pruning rate for each layer based on the performance loss threshold of dynamic iteration and the global pruning rate.**

high-frequency part may contain noise or redundant information, retaining the low-frequency part helps reduce or eliminate unnecessary details, improving the robustness and generalization ability of the features.

Subsequently, we quantify the CSD of the low-frequency part of the gradient enhanced feature map by calculating its L2 norm:

$$\text{CSD}(C_i^l) = \|\text{Rec}(\mathcal{F}_{\text{Low}}(C_i^l), 0)\|_2, \tag{8}$$

where $\|\cdot\|_2$ denotes the L2 norm.

A higher value of CSD implies that the gradient enhanced feature map may carry crucial information for model decision-making, and the corresponding channel is worth retaining.

### 3.3 Adaptive the Pruning Rate for Each Layer

The adaptive pruning algorithm dynamically computes the pruning rate for each convolutional layer and adjusts the pruning strategy based on the model's performance, thus achieving more precise and efficient model compression, as shown in **Figure 4**.

Initially, we decompose the global threshold for performance degradation into progressive descent constraints, represented by the formula:

$$\prod_{l=1}^{L}(1 + d_1\lambda^{l-1}) = \alpha, \tag{9}$$

where $d_1$ represents the initial loss threshold for the first layer (indicating the maximum acceptable performance loss for pruning the first layer), $\alpha$ is the threshold for global performance degradation, and $\lambda$ represents a constant scaling factor for each step based on

---

**Algorithm 1** CSD Pruning Framework

**Input**: Pre-trained weight tensor $W^l$, and desired total pruning rate $P_t$.

**Output**: Pruned weight tensor $W_{\text{prune}}^l$.

1: Initialize $T_{\text{new}} = T$, $k_l = 0$, $P_t = 0.4$.
2: **for** each input sample **do**
3:     Calculate class scores $S_c$ by Equation (2).
4:     **while** $T_{\text{new}} < T \times P_t$ **do**
5:         **for** $i = 1$ to $c^l$ **do**
6:             **while** $\delta_L < d_i$ and $k_l < n_l - 1$ **do**
7:                 Calculate Gradient Enhanced Maps by Eqs. (3)-(5).
8:                 Calculate CSD by Eqs. (6)-(8).
9:                 Select $k_l$ filters with lowest CSD to prune.
10:                Evaluate the model loss $L$ and $\delta_L$, and increment $k_l$.
11:            **end while**
12:            Calculate pruning rate $p_l$.
13:            Update $d_{i+1}$ by Eq. (10).
14:        **end for**
15:        Calculate $T_{\text{new}}$.
16:    **end while**
17: **end for**
18: **Fine-tuning**: Obtain final $W_{\text{prune}}^l$ via fine-tuning $W^l$ with removing the pruned filter channels.
19: **return** Pruned weight tensor $W_{\text{prune}}^l$.

---

the previous threshold. With this, we can compute the dropout threshold $d_l$ for the $l$-th convolutional layer:

$$d_l = \lambda \times d_{(l-1)}, \tag{10}$$

Then, for the $l$-th layer, the desired number of filters $k_l$ to be pruned can be calculated as follows:

$$\max_{p_l} \left\{ \sum_{i=1}^{n_l} GC(C_i^l) - \min\left\{ \sum_{i=1}^{k_l} GC(C_i^l) \right\} \right\}, \tag{11}$$
$$\text{s.t. } \Delta L \leq d_l, \quad k_l \leq n_l - 1,$$

where $n^l$ represents the number of filters in the $l$-th layer, and $k_l$ denotes the number of filters with the minimum global contribution $GC(C_i^l)$ in the $l$-th layer. By sorting the global contribution $GC(C_i^l)$ in the $l$-th layer and selecting a set of filters with the minimum global contribution $GC(C_i^l)$, we find the maximum value of $k_l$ under the constraint of performance loss $\Delta L \leq d_l$. This approach allows us to retain filters that contribute more to performance while controlling the extent of pruning by limiting performance loss.

In each layer, we compute the global contribution of each filter, select and prune the filters with the lowest scores, and then evaluate the model's performance after pruning. Based on the degree of performance degradation, we dynamically adjust the pruning strategy until the specified pruning rate and performance degradation threshold are reached.

## 4 EXPERIMENTS

### 4.1 Implementation Details

**Datasets.** We selected three classification datasets to evaluate the performance of our method: 1) CIFAR-10 [24]; 2) CIFAR-100 [24];

Table 1: Experimental results on CIFAR-10 dataset.

| Model | Algorithm | Baseline(%) | Top-1 Acc.(%) | ΔTop-1 Acc.(%) | FLOPs(↓) | Params(↓) |
|---|---|---|---|---|---|---|
| ResNet-32 | LCCL [6] | 92.33 | 90.74 | -1.59 ↓ | 31.2% | N/A |
| | SFP [17] | 92.63 | 92.08 | -0.55 ↓ | 41.5% | N/A |
| | TAS [7] | 93.89 | 93.16 | -0.73 ↓ | 49.4% | N/A |
| | FPGM [26] | 92.63 | 92.31 | +0.32 ↓ | 41.5% | N/A |
| | DCPH [5] | 93.34 | 92.85 | -0.49 ↑ | 30% | N/A |
| | **Ours** | 92.40 | **93.18** | **+0.78 ↑** | 53.1% | 44.3% |
| ResNet-56 | FTWT [8] | 93.66 | 92.63 | -1.03 ↓ | 60% | N/A |
| | SFP [17] | 93.59 | 93.35 | -0.24 ↓ | 50% | N/A |
| | CFDP [23] | 93.26 | 93.97 | +0.71 ↑ | 28% | 22.3% |
| | LRMF [57] | 93.59 | 93.29 | -0.3 ↓ | 52.6% | N/A |
| | FPGM [26] | 93.59 | 92.93 | -0.66 ↓ | 52.6% | N/A |
| | HRank [31] | 93.26 | 93.85 | +0.59 ↑ | 28% | 22.3% |
| | DCP [16] | 93.80 | 93.49 | -0.31 ↓ | 50% | 49% |
| | **Ours** | 93.26 | **94.16** | **+0.90 ↑** | 51.3% | 44.3% |
| ResNet-110 | HRank [31] | 93.50 | 94.23 | +0.73 ↑ | 41.2% | 39.4% |
| | GAL [33] | 93.50 | 92.55 | -0.95 ↓ | 48.5% | 44.8% |
| | EPruner [30] | 93.50 | 94.23 | +0.73 ↑ | 41.2% | 39.4% |
| | APIB [14] | 93.26 | 93.92 | +0.66 ↑ | 54% | 50% |
| | FalCon [52] | 93.68 | 93.79 | +0.11 ↑ | 60.3% | N/A |
| | FTWT [8] | 93.26 | 92.63 | -0.63 ↓ | 66% | N/A |
| | **Ours** | 93.26 | **94.43** | **+1.17 ↑** | 51.9% | 46.1% |
| VGG-16 | HRank [31] | 93.96 | 93.43 | -0.53 ↓ | 53.5% | 82.9% |
| | GAL [33] | 93.96 | 93.42 | -0.54 ↓ | 45.2% | 82.2% |
| | FPGM [18] | 93.58 | 93.23 | -0.35 ↓ | 35.9% | N/A |
| | FalCon [52] | 93.32 | 91.92 | -1.40 ↓ | 67.3% | N/A |
| | SSS [21] | 93.96 | 93.02 | -0.94 ↓ | 41.6% | 73.8% |
| | **Ours** | 93.96 | **93.69** | **-0.27 ↓** | 45.4% | 47.6% |

3) ImageNet [42]. CIFAR-10 comprises 10 categories such as airplanes, birds, and cats. It consists of 50,000 training images (32×32 pixels) and 10,000 testing images (32×32 pixels). CIFAR-100 is an extension of the CIFAR-10 dataset, featuring 100 classes instead of 10. ImageNet, on the other hand, encompasses over 14 million images, spanning diverse categories from animals and plants to everyday objects, with a total of 20,000 classes. The training images in ImageNet have a resolution of 224×224 pixels.

**Evaluation Metrics.** To assess the model's performance accurately, we adopted three commonly used metrics based on the SOTA: Top-1%, FLOPs and Params. Top-1% reflects the model's recognition ability for the most probable class on a specific dataset. Params and FLOPs evaluate the model's size and computational requirements, respectively. For the ImageNet dataset, due to its difficulty, we included the commonly used Top-5% accuracy as an evaluation metric to comprehensively measure the model's classification performance.

**Configuration.** Our method is implemented with Pytorch. During the training process, a uniform preprocessing of the datasets was applied. For the CIFAR dataset, standard data augmentation techniques such as random scaling, cropping, and rotation were employed. This ensured that the size of the images used for training was uniformly set to 32×32×3. Specifically, the ResNet-32, -56, and -110 are trained for 300 epochs of fine-tuning with a batch size of 218. The momentum is 0.9, the weight decay is 0.005, and the initial learning rate is 0.01. To determine the importance of each filter, 5 batches (640 input images) were randomly sampled to calculate the CSD of each gradient-enhanced feature map in all experiments. For ImageNet and ResNet-50 is trained for 100 epochs with batch size of 256, weight decay of 1e-4, and momentum of 0.9. After pruning, the pruned model was fine-tuned using Stochastic Gradient Descent (SGD) as the optimizer on 8 NVIDIA A100-SXM GPUs.

## 4.2 CIFAR-10 Results

In **Table 1**, we evaluated our method on the CIFAR-10 dataset in both single branch networks (VGGNet) and multi branch networks (ResNet), and compared it with existing pruning methods.

For the ResNet-32 model, our method showed significant advantages compared to other pruning algorithms[5–7, 17, 26]. Our method achieved a 0.78% improvement in Top-1 accuracy, and reduced FLOPs and parameter count by 53.1% and 44.3%, respectively.

For the ResNet-56 model, our CSD prunig approach achieves an accuracy improvement of 0.90% compared to the baseline model. Simultaneously, the number of parameters and FLOPs is reduced by 44.3% and 51.3%, respectively, highlighting the excellent compression performance of this method. Additionally, we compare the experimental results with other methods such as HRank [31], LRMF [57], FTWT [8], and CFDP [23], our approach not only maintains model performance but also further improves predictive accuracy, as shown in **Table 1**.

**Table 2: Experimental results on CIFAR-100 dataset.**

| Model | Algorithm | Top-1(%) | Δ(%) | FLOPs(↓) |
|---|---|---|---|---|
| ResNet-32 | SFP [17] | 68.37 | -0.24 | 50% |
|  | TAS [7] | 72.41 | -0.18 | 38.5% |
|  | LCCL [6] | 67.39 | -0.24 | 50% |
|  | FPGM [26] | 68.52 | -0.66 | 52.6% |
|  | DCPH [5] | 69.51 | -0.31 | 50% |
|  | **Ours** | **71.28** | **+0.13** | 50% |
| ResNet-56 | SFP [17] | 68.79 | +2.61 | 52.6% |
|  | FPGM [26] | 69.66 | +1.75 | 52.6% |
|  | DCPH [5] | 71.31 | -0.41 | 30% |
|  | **Ours** | **71.72** | **+2.03** | 51.5% |
| ResNet-110 | SFP [17] | 71.28 | +2.86 | 52.3% |
|  | FPGM [26] | 72.55 | +1.59 | 52.3% |
|  | DCPH [5] | 72.79 | -0.26 | 30% |
|  | **Ours** | **72.97** | **+1.71** | 51.9% |

For the ResNet-110 model, our algorithm demonstrates superior performance in accuracy compared to other methods, achieving a remarkable 94.43%, significantly outperforming algorithms such as HRank [31], GAL [33], EPruner [30], and APIB [14], as shown in **Table 1**. This implies that our method excels in preserving predictive performance on the pruned model. Relative to the baseline model, our algorithm reduces the parameter count by 46.1% and FLOPs by 51.9%. This highlights the outstanding effectiveness of our method in model compression, resulting in a more lightweight pruned model suitable for resource-constrained environments.

Finally, we validated the performance of our proposed method using VGG-16, as shown in **Table 1**. Not surprisingly, our method demonstrated superior performance in deep compression. Compared to HRank [31], GAL [33], FPMG [18], FalCon [52], and SSS [21], our method exhibited significant advantages in terms of Top-1 accuracy. Specifically, our approach achieved up to a 45.4% reduction in FLOPs (from 93.96% to 93.69%) with only a 0.27% accuracy loss. Additionally, despite achieving a similar reduction in FLOPs, our method incurred less Top-1 accuracy loss compared to FPGM (-0.27% vs. -0.35%). These results underscore the effectiveness of our proposed method.

Overall, our method has better generalization ability and model compression performance on the CIFAR-10 dataset.

## 4.3 CIFAR-100 Results

For the ResNet-32 model, our method achieved significant performance improvement. After pruning, our model achieved a Top-1 accuracy of 71.28%, an increase of 0.13% compared to the baseline, as shown in **Table 2**. This result outperforms other pruning methods (such as SFP and FPGM) and maintains a high pruning rate (50%). This indicates that our method effectively reduces the computational burden of the model while maintaining performance.

For the ResNet-56 model, our method also yielded satisfactory results. After pruning, our model achieved a top-1 accuracy of 71.72%, an improvement of 2.03% compared to the baseline. Compared to other methods, our method demonstrates a more significant improvement in accuracy while still achieving a high pruning rate

(51.5%). This suggests that our method has a performance advantage, even on larger models.

In the case of the ResNet-110 model, our method also delivered notably significant results. Our model achieved a top-1 accuracy of 72.97%, an increase of 1.71% compared to the baseline. Compared to other pruning methods, our approach shows competitiveness in accuracy improvement while maintaining a high pruning rate (51.9%). This further demonstrates the effectiveness of our method across models of various sizes.

Overall, our method consistently demonstrates superior performance in improving model accuracy while maintaining high pruning rates across ResNet models of different depths. These results underscore the effectiveness and versatility of our proposed pruning approach.

## 4.4 ImageNet Results

We tested ResNet-50 on ImageNet. Our algorithm achieves a minimal decrease of 0.04% in Top-1 accuracy, indicating performance close to the original model. The slight reduction in Top-5 accuracy, by only 0.08%, signifies a notable success in preserving the primary predictive performance, as shown in **Table 3**. In comparison to other algorithms such as ABCPruner [32], CPS [53], FalCon [52], APIB [14], and AutoPruner [37], our method demonstrates a well-balanced performance in maintaining accuracy and compressing the model. This highlights superior overall effectiveness.

## 4.5 Ablation

In this section, we delve into the motivations behind the CSD pruning design choices. Specifically, we examine how each component of our framework influences the overall performance of our approach. For consistency, we conduct all ablation studies on the CIFAR-10 dataset using ResNet-56.

*4.5.1 Gradient Enhanced Feature Map.* We investigate the impact of the Gradient Enhanced Feature Map (G-E Feature Map) on the CSD pruning method. The G-E Feature Map is crucial for understanding channel importance. We compared the different performances of channel importance guided pruning using G-E feature map ensemble and feature map set calculation, as shown in **Table 4**. In the presence of DWT, enabling G-E feature map further enhances the model performance, resulting in a Top-1 accuracy of 94.18%. Compared to using DWT alone without enabling G-E feature map, the accuracy improves by 0.76% (93.72% -> 94.18%). Enabling G-E feature map also brings additional reductions in FLOPs and parameters, by 40% and 45%, respectively, as shown in **Table 4**. This indicates that G-E feature map play a positive role in further compressing the model's computational burden and parameter count.

*4.5.2 Enhancement of Low-Level Features.* We assess the significance of the DWT step in the CSD pruning method. We compare the pruning performance with and without the DWT step. When DWT is applied, the CSD pruning method shows a significant improvement in Top-1 accuracy, reaching 94.18%, an increase of 0.16% (94.02% -> 94.18%) points compared to the scenario without DWT. Simultaneously, under the usage of DWT, there is a 40% reduction in FLOPs and a 45% reduction in parameters, as shown in **Table 4**.

**Table 3: Experimental results of ResNet-50 on ImageNet dataset.**

| Algorithm | Top-1 Acc.(%) | ΔTop-1 Acc.(%) | Top-5 Acc.(%) | ΔTop-5 Acc.(%) | FLOPs(↓) | Params(↓) |
|---|---|---|---|---|---|---|
| HRank [31] | 76.15→74.98 | -1.17 ↓ | 92.87→92.33 | -0.54 ↓ | 43.7% | 36.6% |
| GAL [33] | 76.15→71.95 | -4.20 ↓ | 92.96→90.79 | -2.17 ↓ | 43.7% | 36.6% |
| ABCPruner [32] | 76.01→73.86 | -2.15 ↓ | 92.96→91.69 | -1.27 ↓ | 54.3% | N/A |
| DCP [60] | 76.01→74.95 | -1.06 ↓ | 92.93.15→92.32 | -0.61 ↓ | 55.76% | N/A |
| CPS [53] | 76.15→75.59 | -0.56 ↓ | N/A | N/A | 44.3% | N/A |
| MFP 30% [19] | 76.15→75.67 | -0.48 ↓ | 92.87→92.81 | -0.06 ↓ | 42.2% | N/A |
| FalCon [52] | 75.83→74.59 | -1.24 ↓ | 92.78→92.51 | -0.27 ↓ | 53.5% | N/A |
| APIB [14] | 76.15→76.07 | -1.24 ↓ | N/A | N/A | 56% | 50% |
| SASL [45] | 76.15→75.15 | -1.00 ↓ | 92.87→92.58 | -0.40 ↓ | 56.1% | 50% |
| TRP [51] | 75.90→72.69 | -3.21 ↓ | 92.70→91.41 | -1.29 ↓ | 56.52% | N/A |
| AutoPruner [37] | 76.15→74.76 | -1.39 ↓ | 92.87→92.15 | -0.72 ↓ | 51.2% | N/A |
| **Ours** | 76.15→76.11 | **-0.04 ↓** | 92.87→92.79 | **-0.08 ↓** | 53.7% | 40.8% |

**Table 4: Ablation results on CSD pruning.**

| DWT | Maps | Top-1(%) | FLOPs(↓) | Params(↓) |
|---|---|---|---|---|
| ✓ | G-E Feature Map | 94.18 | 40% | 45% |
| - | G-E Feature Mas | 94.02 | 40% | 45% |
| ✓ | Feature Map | 93.72 | 40% | 45% |
| - | Feature Map | 93.64 | 40% | 45% |

**Table 5: Impact of adaptive pruning rate designs.**

| Adptive | Pruning rate | Top-1(%) | FLOPs(↓) | Paramrs(↓) |
|---|---|---|---|---|
| ✓ | 35(%) | 94.21 | 35% | 34% |
| - | 35(%) | 93.24 | 35% | 34% |
| ✓ | 40(%) | 94.17 | 41% | 40% |
| - | 40(%) | 93.38 | 40% | 39% |
| ✓ | 50(%) | 94.16 | 51% | 48% |
| - | 50(%) | 93.33 | 50% | 47% |

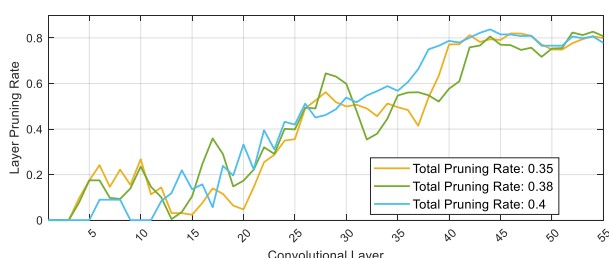

**Figure 5: Effect of adaptive pruning rate designs on CSD. We compare the adaptive trend of pruning rates for each layer when different and relatively close total pruning rates (0.35, 0.38, 0.4) are set.**

This indicates that DWT, while enhancing model accuracy, effectively reduces the computational burden and parameter count of the model.

*4.5.3 Adaptive Pruning Rate.* We explore different designs for adaptive pruning rates in the CSD pruning method. The choice of pruning rate calculation method significantly influences the overall pruning performance, comparing performance with and without adaptive pruning rate designs. When adaptive pruning rate designs were enabled, we observed variations in model performance under different pruning rates $p_t$. Taking a 35% pruning rate as an example, the Top-1 accuracy reached 94.21%. Compared to the scenario without adaptive pruning rate designs, FLOPs were reduced by 35%, and parameters decreased by 34%. We further validated performance at 40% and 50% pruning rates, as shown in **Table 5**. In both cases, the model achieved higher accuracy while achieving more significant reductions in FLOPs and parameters. This indicates that adaptive pruning rate designs contribute to optimizing model performance under different pruning rates, leading to more effective model compression.

In addition, we observed that when different batches of images are used as input, but similar or identical total pruning rates are set,

the adaptive mechanism calculates approximately equal pruning rates for each layer, as shown in **Figure 5**. This indicates that the model maintains a consistent pruning trend for each layer under various input conditions. This aligns with our initial hypothesis, where shallow layers exhibit high spatial dependencies, resulting in smaller pruning rates, while deep layers with lower spatial dependencies have larger pruning rates.

These ablation studies provide insights into the importance of each component in the CSD pruning framework, guiding future applications and improvements.

## CONCLUSION

In this paper, we introduced CSD pruning, a novel pruning strategy that adaptively calculates pruning rates for each layer, making the pruning process more intelligent. Our algorithm achieves high accuracy on CIFAR-10, CIFAR-100 and ImageNet while significantly reducing the model's size and computational complexity, showcasing the exceptional performance of our approach. Furthermore, we conducted several ablation studies demonstrating the robustness of each proposed component to the initialization of the model. Overall, we have demonstrated the true advantages of our framework and evaluation criteria against benchmark standards. In future work, we aim to delve deeper into theoretical analyses on how interpretability can be combined with channel prunability.

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
