# OpenReview forum: "Adaptive Pruning of Channel Spatial Dependability in Convolutional Neural Networks"
_acmmm.org/ACMMM/2024/Conference — MM2024 Poster_

### Official Review · Reviewer_hDPG · 2024-05-24

**Rating:** 2
**Confidence:** 3

**Summary:**

This paper focuses on CNN model pruning and establishes a description of the contribution of channels based on Discrete Wavelet Transform (DWT), referred to as the Channel Spatial Dependability (CSD) metric. It also presents an adaptive pruning strategy.

**Strengths:**

1. Evaluating the contribution of channels based on Discrete Wavelet Transform (DWT) is novel for me.

2. The experiments are comprehensive.

**Limitations:**

1. The paper focuses on pruning CNNs, yet CNNs are not only applied to classification tasks. From Equation 2 and related content, it seems that the operations described in the paper are limited to classification tasks, leading me to question the generalizability of this method. It seems that a more rigorous description of gradient calculation is necessary, rather than solely deriving conclusions from classification results. This is a part where the authors need to make improvements.

2. The abstract mentions that existing pruning methods "overlook its 'interpretability' information" and that "we explore CNN pruning strategies from the perspective of model interpretability." However, to my knowledge, this paper merely evaluates channel importance based on the Discrete Wavelet Transform (DWT), while existing methods have various ways to assess this importance, such as rank, similarity, and gradient. These methods, including the one proposed in this paper, are valid to some extent. Nonetheless, I do not agree that the proposed method is inherently interpretable, as DWT cannot be proven to be the sole metric for evaluating channel importance (if the authors consider DWT interpretable in terms of importance, then other methods could also be regarded as interpretable).

3. The overall writing quality needs improvement. There are many minor errors in both the figures and the text, such as "map map" in line 505 and "Feature Mas" in line 833. Additionally, in Figure 3, there is a "GF" label on the left side, but I couldn't find its definition anywhere in the paper. Furthermore, the structure of Figure 3 seems to require enhancement; it appears that the forward and backward processes of the network could be illustrated in a single area. The tables contain too many instances of "N/A," and it is recommended to standardize the decimal places for "FLOPs" (some are integers, while others have one decimal place).

**Suitability:**

2

---

### Official Review · Reviewer_rHQT · 2024-05-25

**Rating:** 3
**Confidence:** 3

**Summary:**

This paper presents a new metric called CSD (Channel Spatial Dependability), which is used to rank the importance of convolutional filters. The main idea is to retain filters with high global contribution towards the network's performance, and remove redundant filters to obtain lightweight CNNs.

**Strengths:**

-- This work is a good step towards the evolving research on model compression for resource-constrained execution. Authors motivate that neural networks are overparametrized, hence its an important scientific question to obtain small models with considerable accuracy.

-- The main innovation made by authors appears to be in the form of CSD metric to evaluate filter importance.

-- In particular, authors apply wavelet transform on feature maps to obtain low and high frequency components. Then, l2-norm over low-frequency components is used to decide on high and less important features. The idea is interesting, with an integration of adapting layer-wise sparsities as the pruning is carried out.

--Overall, the paper is well motivated and contributions are clearly presented.

**Limitations:**

--While overall write-up is good, there is still room for improvement. There are several typos, for example, in Table 5, '' Adptive -> Adaptive ''.

-- Similar inconsistencies exist across the paper, for example, in caption, "nl" is referred as maximum compression ratio, but in the text, it's again associated to other terms like the number of filters and output channels, causing confusion to readers. Again, in the text, the final compression ratio for layer L is written as "nl_new" and which should be consistent across text and caption too.

-- It is also confusing whether authors are trying to prune 'channels' or 'filters', because at some instance, 'channels' are pruned, while 'filters' at others. Please see line 136 and 351 as an example.

--It looks like authors have not established a clear difference between channels and filters yet. There seems to be a gap in understanding both of these important structural components of CNNs as removal of either may have different impact on overall architecture, performance and complexity.

-- Similarly, line 98 in Intro should be more clarified as to what structural changes are evident in structured and unstructured pruning. I think change happens across both methods, but what authors mentioned seems to be contrasting.

-- It would be also important to see how does the performance varies if pruning rates are relaxed, and allowed to go beyond 0.5. I assume a severe degradation in performance which might question the effectiveness of the proposed DWT based importance estimation.

-- The biggest concern I have is that while pruning aims to relieve the overall computational burden of CNNs, the proposed pruning pipeline seems to introduce additional computational overhead by calculating complex wavelet coefficients and then projecting L2-norm over each component. There needs to be well established reasons as to why these computationally complex steps are necessarily taken? Whether they help in retaining accuracy for extremely compressed models?

-- Finally, it would be interesting to see how effective is the proposed pruning is for already mobile-optimized models such as MobileNets or EfficientNets.

**Suitability:**

2

---

### Official Review · Reviewer_dP9v · 2024-05-27

**Rating:** 4
**Confidence:** 2

**Summary:**

This paper presents a method for structured CNN pruning aiming for model compression on resource-constrained devices. This method improves upon traditional pruning strategies by focusing on model interpretability and the internal structure of CNNs, using class saliency maps and Discrete Wavelet Transform (DWT) to assess the importance of channels within the model without being affected by high frequency noise. This paper also proposed an adaptive approach to dynamically adjust the pruning rates layer by layer based on performance metrics, leading to significant size and computation reduction while preserving or even improving model accuracy. This paper also provided extensive experiments and ablation study to verify the effectiveness of the proposed method.

**Strengths:**

1. Unlike other methods that prune CNNs using norms or shapes, this paper introduces a new way that uses gradient-enhanced feature maps. This method provides a clearer view of how gradients affect model pruning and helps us better understand the importance of each channel.

2. This method also uses a Discrete Wavelet Transform module to remove high-frequency noise when assessing the importance of channels and filters. This improvement makes the pruning process more reliable.

3. The paper describes an adaptive pruning strategy that adjusts to changes in channel importance as pruning happens. This flexible approach ensures that the model remains accurate while becoming simpler.

4. The overall presentation is clear and effective.

**Limitations:**

1. The method is designed only for CNNs used in classification tasks, which limits its wider application. It would be more useful if the author could extend this approach to work with other types of neural networks, like transformers, and for different tasks beyond classification.

2. The explanation of how pruning rates are adjusted for each layer needs to be clearer. It's unclear how the parameters $\lambda$, $d_i$, and $\alpha$ are chosen when the only given input is the overall pruning rate $p$.

3. A basic concern is that the main goal is to reduce the number of floating-point operations (FLOP). However, just having fewer FLOPs doesn’t always mean a model will run more efficiently on devices with limited resources (like being faster than a model with more FLOPs). This assumption should be examined more closely to confirm if minimizing FLOPs is the best way to define the problem.

**Suitability:**

3

---

### Official Review · Reviewer_RhJp · 2024-06-04

**Rating:** 5
**Confidence:** 3

**Summary:**

This paper proposed a novel channel pruning procedure for convolutional neural networks focusing on taking the advantage of the interpretable information corresponding to each channel for accurate measurement of their importance. An end-to-end framework has been presented for prunning off less contributive channels of any 2D convolutional layers that distinctively incorporating multiple techniques namely gradient-enhanced feature map, DWT filter criterion, and layered adaptive pruning rate. Extensive experiments are covered in this study which provides all-round evidence that the proposed method is significantly better than state of arts in accuracy preservation and compression efficiency.

**Strengths:**

1. *Impressive experimental results and gain from the state of arts.* The contribution of the proposed method is reasonably validated by its consistently better accuracy preservation performance across all the dimensions of the experiment setup, including dataset, network architecture, and compresssion rate.

2. *The proposed idea has solid novelty and provided sound rationale behind.* As the interpretability characteristics of machine learning models have been broadly studied recently, making it benefit model compression techniques are naturally a promissing direction.

3. *High presentation quality.* The overall writing of this draft is at a high standard. Except the minor artifacts to be resolved (as pointed out below), this draft is already in a feasible state for publishing.

**Limitations:**

1. *Lack of perceptual results and analysis.* Sinc the proposed idea is stacked upon the saliency map based criterion, it is surprising to readers when we are not shown the actual saliency map or similar visualization of the experiemented model internals at all. It would be strong evidence to support that adopting the gradient-enhanced feature maps has a key contribution to the end pruning effect improvement if the difference can be demonstrated intuitively. Is the infusion of interpretability information perceivable from the feature map
to explain it would be a better way to guide the selection of pruned channel?

2. *Experimetal flaws.*
    1. In the ablation study, it is claimed enabling G-E feature map brings additional reduction in FLOPS. This is not a straightforward conclusion to draw and authors are suggested to add some derivational content to explain how this is the case.
    2. As shown in Table 4, the application of DWT has minimal effect on the accuracy gain. Besides, looking from the implementation details, this is an independent technique from CSD. It should be elaborated how DWT matters to this work and how it is related to completeness of the proposed idea.
    3. For the experiment on ImageNet, comparing the accuracy drop to [19], it is interesting to see the Top-5 accuracy drop is even higher, contradicting to Top-1 accuracy result. Any observations to why?
    4. No runtime evaluation for the efficiency of the pruning procedure, as compared to other methods, and relative to the plain training job.

3. *Writing artifacts and nitpicking.*
    1. Abstract, "quantization pruning" -> "pruning".
    2. Introduction, "In summary, while these ..."
    3. Section 2.1, "who suggest ..." -> "which suggest ..."
    4. Figure 3, the caption is too abstract and should be focusing on explaining the quoted figure
    5. Section 3.3, the title, "Adaptive the Pruning Rate for Each Layer", remove "the"

**Suitability:**

3

---

### Meta-Review · Area_Chair_xHeb · 2024-07-01

**Recommendation:** Accept (Poster)
**Confidence:** 4

**Metareview:**

This paper presents a novel CNN pruning method using Channel Spatial Dependability (CSD) metric and an adaptive pruning strategy. Some concerns were raised about generalizability beyond classification tasks and the computational overhead of the method, the overall consensus is that the paper makes a significant contribution to model compression research. Given the paper's novelty, solid rationale, and impressive experimental results, I recommend accepting this submission with the condition that the authors address the minor writing issues and clarify some technical details in the final version.